# Sparse Local Embeddings for Extreme Multi-label Classification

**Kush Bhatia[†], Himanshu Jain[§], Purushottam Kar[‡*], Manik Varma[†], and Prateek Jain[†]**
[†]Microsoft Research, India
[§]Indian Institute of Technology Delhi, India
[‡]Indian Institute of Technology Kanpur, India
{t-kushb,prajain,manik}@microsoft.com
himanshu.j689@gmail.com, purushot@cse.iitk.ac.in

## Abstract

The objective in extreme multi-label learning is to train a classifier that can automatically tag a novel data point with the most relevant *subset* of labels from an extremely large label set. Embedding based approaches attempt to make training and prediction tractable by assuming that the training label matrix is low-rank and reducing the effective number of labels by projecting the high dimensional label vectors onto a low dimensional linear subspace. Still, leading embedding approaches have been unable to deliver high prediction accuracies, or scale to large problems as the low rank assumption is violated in most real world applications.

In this paper we develop the SLEEC classifier to address both limitations. The main technical contribution in SLEEC is a formulation for learning a small ensemble of local distance preserving embeddings which can accurately predict infrequently occurring (tail) labels. This allows SLEEC to break free of the traditional low-rank assumption and boost classification accuracy by learning embeddings which preserve pairwise distances between only the nearest label vectors.

We conducted extensive experiments on several real-world, as well as benchmark data sets and compared our method against state-of-the-art methods for extreme multi-label classification. Experiments reveal that SLEEC can make significantly more accurate predictions then the state-of-the-art methods including both embedding-based (by as much as 35%) as well as tree-based (by as much as 6%) methods. SLEEC can also scale efficiently to data sets with a million labels which are beyond the pale of leading embedding methods.

## 1 Introduction

In this paper we develop SLEEC (Sparse Local Embeddings for Extreme Classification), an extreme multi-label classifier that can make significantly more accurate and faster predictions, as well as scale to larger problems, as compared to state-of-the-art embedding based approaches.

eXtreme Multi-label Learning (**XML**) addresses the problem of learning a classifier that can automatically tag a data point with the most relevant *subset* of labels from a large label set. For instance, there are more than a million labels (categories) on Wikipedia and one might wish to build a classifier that annotates a new article or web page with the subset of most relevant Wikipedia categories. It should be emphasized that multi-label learning is distinct from multi-class classification where the aim is to predict a single mutually exclusive label.

**Challenges**: XML is a hard problem that involves learning with hundreds of thousands, or even millions, of labels, features and training points. Although, some of these problems can be ameliorated

---

[*]This work was done while P.K. was a postdoctoral researcher at Microsoft Research India.

using a label hierarchy, such hierarchies are unavailable in many applications [1, 2]. In this setting, an obvious baseline is thus provided by the *1-vs-All* technique which seeks to learn an an independent classifier per label. As expected, this technique is infeasible due to the prohibitive training and prediction costs given the large number of labels.

**Embedding-based approaches**: A natural way of overcoming the above problem is to reduce the effective number of labels. Embedding based approaches try to do so by projecting label vectors onto a low dimensional space, based on an assumption that the label matrix is low-rank. More specifically, given a set of $n$ training points $\{(\mathbf{x}_i, \mathbf{y}_i)_{i=1}^n\}$ with $d$-dimensional feature vectors $\mathbf{x}_i \in \mathbb{R}^d$ and $L$-dimensional label vectors $\mathbf{y}_i \in \{0, 1\}^L$, state-of-the-art embedding approaches project the label vectors onto a lower $\widehat{L}$-dimensional linear subspace as $\mathbf{z}_i = \mathbf{U}\mathbf{y}_i$. Regressors are then trained to predict $\mathbf{z}_i$ as $\mathbf{V}\mathbf{x}_i$. Labels for a novel point $\mathbf{x}$ are predicted by post-processing $\mathbf{y} = \mathbf{U}^\dagger \mathbf{V}\mathbf{x}$ where $\mathbf{U}^\dagger$ is a decompression matrix which lifts the embedded label vectors back to the original label space.

Embedding methods mainly differ in the choice of their compression and decompression techniques such as compressed sensing [3], Bloom filters [4], SVD [5], landmark labels [6, 7], output codes [8], *etc*. The state-of-the-art LEML algorithm [9] directly optimizes for $\mathbf{U}^\dagger$, $\mathbf{V}$ using a regularized least squares objective. Embedding approaches have many advantages including simplicity, ease of implementation, strong theoretical foundations, the ability to handle label correlations, as well as adapt to online and incremental scenarios. Consequently, embeddings have proved to be the most popular approach for tackling XML problems [6, 7, 10, 4, 11, 3, 12, 9, 5, 13, 8, 14].

Embedding approaches also have limitations – they are slow at training and prediction even for small embedding dimensions $\widehat{L}$. For instance, on WikiLSHTC [15, 16], a Wikipedia based challenge data set, LEML with $\widehat{L} = 500$ takes $\sim 12$ hours to train even with early termination whereas prediction takes nearly 300 milliseconds per test point. In fact, for text applications with $\widehat{d}$-sparse feature vectors such as WikiLSHTC (where $\widehat{d} = 42 \ll \widehat{L} = 500$), LEML's prediction time $\Omega(\widehat{L}(\widehat{d} + L))$ can be an order of magnitude more than even 1-vs-All's prediction time $O(\widehat{d}L)$.

More importantly, the critical assumption made by embedding methods, that the training label matrix is low-rank, is violated in almost all real world applications. Figure 1(a) plots the approximation error in the label matrix as $\widehat{L}$ is varied on the WikiLSHTC data set. As is clear, even with a 500-dimensional subspace the label matrix still has 90% approximation error. This happens primarily due to the presence of hundreds of thousands of "tail" labels (Figure 1(b)) which occur in at most 5 data points each and, hence, cannot be well approximated by any linear low dimensional basis.

**The SLEEC approach**: Our algorithm SLEEC extends embedding methods in multiple ways to address these limitations. First, instead of globally projecting onto a linear low-rank subspace, SLEEC learns embeddings $\mathbf{z}_i$ which non-linearly capture label correlations by preserving the pairwise distances between only the closest (rather than all) label vectors, i.e. $d(\mathbf{z}_i, \mathbf{z}_j) \approx d(\mathbf{y}_i, \mathbf{y}_j)$ only if $i \in \text{kNN}(j)$ where $d$ is a distance metric. Regressors $\mathbf{V}$ are trained to predict $\mathbf{z}_i = \mathbf{V}\mathbf{x}_i$. We propose a novel formulation for learning such embeddings that can be formally shown to consistently preserve nearest neighbours in the label space. We build an efficient pipeline for training these embeddings which can be orders of magnitude faster than state-of-the-art embedding methods.

During prediction, rather than using a decompression matrix, SLEEC uses a k-nearest neighbour (kNN) classifier in the embedding space, thus leveraging the fact that nearest neighbours have been preserved during training. Thus, for a novel point $\mathbf{x}$, the predicted label vector is obtained using $\mathbf{y} = \sum_{i:\mathbf{V}\mathbf{x}_i \in \text{kNN}(\mathbf{V}\mathbf{x})} \mathbf{y}_i$. The use of a kNN classifier is well motivated as kNN outperforms discriminative methods in acutely low training data regimes [17] as is the case with tail labels.

The superiority of SLEEC's proposed embeddings over traditional low-rank embeddings can be seen by looking at Figure 1, which shows that the relative approximation error in learning SLEEC's embeddings is significantly smaller as compared to the low-rank approximation error. Moreover, we also find that SLEEC can improve the prediction accuracy of state-of-the-art embedding methods by as much as 35% (absolute) on the challenging WikiLSHTC data set. SLEEC also significantly outperforms methods such as WSABIE [13] which also use kNN classification in the embedding space but learn their embeddings using the traditional low-rank assumption.

**Clustering based speedup**: However, kNN classifiers are known to be slow at prediction. SLEEC therefore clusters the training data into $C$ clusters, learning a separate embedding per cluster and performing kNN classification within the test point's cluster alone. This allows SLEEC to be more

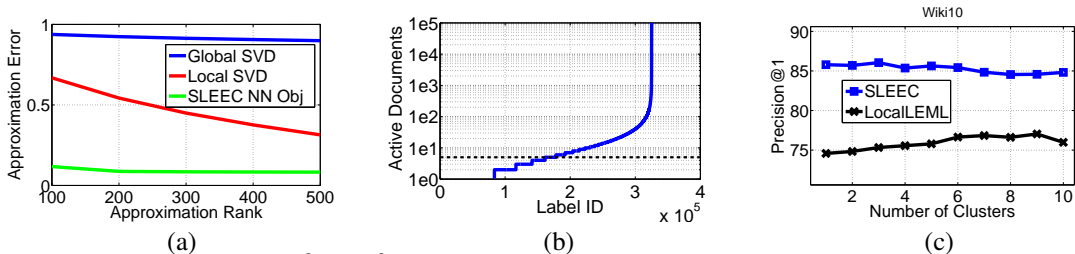

Figure 1: (a) error $\|Y - Y_{\widehat{L}}\|_F^2 / \|Y\|_F^2$ in approximating the label matrix $Y$. Global SVD denotes the error incurred by computing the rank $\widehat{L}$ SVD of $Y$. Local SVD computes rank $\widehat{L}$ SVD of $Y$ within each cluster. SLEEC NN objective denotes SLEEC's objective function. Global SVD incurs $90\%$ error and the error is decreasing at most linearly as well. (b) shows the number of documents in which each label is present for the WikiLSHTC data set. There are about $300K$ labels which are present in $< 5$ documents lending it a 'heavy tailed' distribution. (c) shows Precision@1 accuracy of SLEEC and localLEML on the Wiki-10 data set as we vary the number of clusters.

than two orders of magnitude faster at prediction than LEML and other embedding methods on the WikiLSHTC data. In fact, SLEEC also scales well to the Ads1M data set involving a million labels which is beyond the pale of leading embedding methods. Moreover, the clustering trick does not significantly benefit other state-of-the-art methods (see Figure 1(c)), thus indicating that SLEEC's embeddings are key to its performance boost.

Since clustering can be unstable in large dimensions, SLEEC compensates by learning a small ensemble where each individual learner is generated by a different random clustering. This was empirically found to help tackle instabilities of clustering and significantly boost prediction accuracy with only linear increases in training and prediction time. For instance, on WikiLSHTC, SLEEC's prediction accuracy was $55\%$ with an 8 millisecond prediction time whereas LEML could only manage $20\%$ accuracy while taking 300 milliseconds for prediction per test point.

**Tree-based approaches**: Recently, tree based methods [1, 15, 2] have also become popular for XML as they enjoy significant accuracy gains over the existing embedding methods. For instance, FastXML [15] can achieve a prediction accuracy of $49\%$ on WikiLSHTC using a 50 tree ensemble. However, using SLEEC, we are now able to extend embedding methods to outperform tree ensembles, achieving $49.8\%$ with 2 learners and $55\%$ with 10. Thus, SLEEC obtains the best of both worlds – achieving the highest prediction accuracies across all methods on even the most challenging data sets, as well as retaining all the benefits of embeddings and eschewing the disadvantages of large tree ensembles such as large model size and lack of theoretical understanding.

## 2 Method

Let $\mathcal{D} = \{(\mathbf{x}_1, \mathbf{y}_1) \ldots (\mathbf{x}_n, \mathbf{y}_n)\}$ be the given training data set, $\mathbf{x}_i \in \mathcal{X} \subseteq \mathbb{R}^d$ be the input feature vector, $\mathbf{y}_i \in \mathcal{Y} \subseteq \{0, 1\}^L$ be the corresponding label vector, and $y_{ij} = 1$ iff the $j$-th label is turned on for $\mathbf{x}_i$. Let $X = [\mathbf{x}_1, \ldots, \mathbf{x}_n]$ be the data matrix and $Y = [\mathbf{y}_1, \ldots, \mathbf{y}_n]$ be the label matrix. Given $\mathcal{D}$, the goal is to learn a multi-label classifier $f : \mathbb{R}^d \to \{0, 1\}^L$ that accurately predicts the label vector for a given test point. Recall that in XML settings, $L$ is very large and is of the same order as $n$ and $d$, ruling out several standard approaches such as 1-vs-All.

We now present our algorithm SLEEC which is designed primarily to scale efficiently for large $L$. Our algorithm is an embedding-style algorithm, i.e., during training we map the label vectors $\mathbf{y}_i$ to $\widehat{L}$-dimensional vectors $\mathbf{z}_i \in \mathbb{R}^{\widehat{L}}$ and learn a set of regressors $V \in \mathbb{R}^{\widehat{L} \times d}$ s.t. $\mathbf{z}_i \approx V\mathbf{x}_i, \forall i$. During the test phase, for an unseen point $\mathbf{x}$, we first compute its embedding $V\mathbf{x}$ and then perform kNN over the set $[V\mathbf{x}_1, V\mathbf{x}_2, \ldots, V\mathbf{x}_n]$. To scale our algorithm, we perform a clustering of all the training points and apply the above mentioned procedures in each of the cluster separately. Below, we first discuss our method to compute the embeddings $\mathbf{z}_i$s and the regressors $V$. Section 2.2 then discusses our approach for scaling the method to large data sets.

### 2.1 Learning Embeddings

As mentioned earlier, our approach is motivated by the fact that a typical real-world data set tends to have a large number of tail labels that ensure that the label matrix $Y$ cannot be well-approximated using a low-dimensional linear subspace (see Figure 1). However, $Y$ can still be accurately modeled

**Algorithm 1** SLEEC: Train Algorithm

**Require:** $\mathcal{D} = \{(x_1, y_1) \ldots (x_n, y_n)\}$, embedding dimensionality: $\widehat{L}$, no. of neighbors: $\bar{n}$, no. of clusters: $C$, regularization parameter: $\lambda, \mu$, L1 smoothing parameter $\rho$
1: Partition $X$ into $Q^1, .., Q^C$ using $k$-means
2: **for** each partition $Q^j$ **do**
3:     Form $\Omega$ using $\bar{n}$ nearest neighbors of each label vector $\mathbf{y}_i \in Q^j$
4:     $[U\ \Sigma] \leftarrow \text{SVP}(P_\Omega(Y^j Y^{j^T}), \widehat{L})$
5:     $Z^j \leftarrow U\Sigma^{\frac{1}{2}}$
6:     $V^j \leftarrow ADMM(X^j, Z^j, \lambda,\ \mu,\ \rho)$
7:     $Z^j = V^j X^j$
8: **end for**
9: **Output**: $\{(Q^1, V^1, Z^1), \ldots, (Q^C, V^C, Z^C\}$

---

**Algorithm 2** SLEEC: Test Algorithm

**Require:** Test point: $\mathbf{x}$, no. of NN: $\bar{n}$, no. of desired labels: $p$
1: $Q_\tau$: partition closest to $\mathbf{x}$
2: $\mathbf{z} \leftarrow V^\tau \mathbf{x}$
3: $\mathcal{N}_z \leftarrow \bar{n}$ nearest neighbors of $z$ in $Z^\tau$
4: $P_x \leftarrow$ empirical label dist. for points $\in \mathcal{N}_z$
5: $y_{pred} \leftarrow Top_p(P_x)$

---

**Sub-routine 3** SLEEC: SVP

**Require:** Observations: $G$, index set: $\Omega$, dimensionality: $\widehat{L}$
1: $M_1 := 0, \eta = 1$
2: **repeat**
3:     $\widehat{M} \leftarrow M + \eta(G - P_\Omega(M))$
4:     $[U\ \Sigma] \leftarrow \text{Top-EigenDecomp}(\widehat{M}, \widehat{L})$
5:     $\Sigma_{ii} \leftarrow \max(0, \Sigma_{ii}), \forall i$
6:     $M \leftarrow U \cdot \Sigma \cdot U^T$
7: **until** Convergence
8: **Output**: $U, \Sigma$

---

**Sub-routine 4** SLEEC: ADMM

**Require:** Data Matrix : $X$, Embeddings : $Z$, Regularization Parameter : $\lambda, \mu$, Smoothing Parameter : $\rho$
1: $\beta := 0, \alpha := 0$
2: **repeat**
3:     $Q \leftarrow (Z + \rho(\alpha - \beta))X^\top$
4:     $V \leftarrow Q(XX^\top(1 + \rho) + \lambda I)^{-1}$
5:     $\alpha \leftarrow (VX + \beta)$
6:     $\alpha_i = \text{sign}(\alpha_i) \cdot \max(0, |\alpha_i| - \frac{\mu}{\rho}), \forall i$
7:     $\beta \leftarrow \beta + VX - alpha$
8: **until** Convergence
9: **Output**: $V$

---

using a low-dimensional non-linear manifold. That is, instead of preserving distances (or inner products) of a given label vector to all the training points, we attempt to preserve the distance to only a few nearest neighbors. That is, we wish to find a $\widehat{L}$-dimensional embedding matrix $Z = [\mathbf{z}_1, \ldots, \mathbf{z}_n] \in \mathbb{R}^{\widehat{L} \times n}$ which minimizes the following objective:

$$\min_{Z \in \mathbb{R}^{\widehat{L} \times n}} \|P_\Omega(Y^T Y) - P_\Omega(Z^T Z)\|_F^2 + \lambda\|Z\|_1, \tag{1}$$

where the index set $\Omega$ denotes the set of neighbors that we wish to preserve, i.e., $(i, j) \in \Omega$ iff $j \in \mathcal{N}_i$. $\mathcal{N}_i$ denotes a set of nearest neighbors of $i$. We select $\mathcal{N}_i = \arg\max_{S, |S| \leq \alpha \cdot n} \sum_{j \in S}(\mathbf{y}_i^T \mathbf{y}_j)$, which is the set of $\alpha \cdot n$ points with the largest inner products with $\mathbf{y}_i$. $|\mathcal{N}|$ is always chosen large enough so that distances (inner products) to a few far away points are also preserved while optimizing for our objective function. This prohibits non-neighboring points from entering the immediate neighborhood of any given point. $P_\Omega : \mathbb{R}^{n \times n} \to \mathbb{R}^{n \times n}$ is defined as:

$$(P_\Omega(Y^T Y))_{ij} = \begin{cases} \langle \mathbf{y}_i, \mathbf{y}_j \rangle, & \text{if } (i, j) \in \Omega, \\ 0, & \text{otherwise.} \end{cases} \tag{2}$$

Also, we add $L_1$ regularization, $\|Z\|_1 = \sum_i \|\mathbf{z}_i\|_1$, to the objective function to obtain sparse embeddings. Sparse embeddings have three key advantages: a) they reduce prediction time, b) reduce the size of the model, and c) avoid overfitting. Now, given the embeddings $Z = [\mathbf{z}_1, \ldots, \mathbf{z}_n] \in \mathbb{R}^{\widehat{L} \times n}$, we wish to learn a multi-regression model to predict the embeddings $Z$ using the input features. That is, we require that $Z \approx VX$ where $V \in \mathbb{R}^{\widehat{L} \times d}$. Combining the two formulations and adding an $L_2$-regularization for $V$, we get:

$$\min_{V \in \mathbb{R}^{\widehat{L} \times d}} \|P_\Omega(Y^T Y) - P_\Omega(X^T V^T V X)\|_F^2 + \lambda\|V\|_F^2 + \mu\|VX\|_1. \tag{3}$$

Note that the above problem formulation is somewhat similar to a few existing methods for non-linear dimensionality reduction that also seek to preserve distances to a few near neighbors [18, 19]. However, in contrast to our approach, these methods do not have a direct out of sample generalization, do not scale well to large-scale data sets, and lack rigorous generalization error bounds.

**Optimization**: We first note that optimizing (3) is a significant challenge as the objective function is non-convex as well as non-differentiable. Furthermore, our goal is to perform optimization for data

sets where $L, n, d \gg 100,000$. To this end, we divide the optimization into two phases. We first learn embeddings $Z = [\mathbf{z}_1, \ldots, \mathbf{z}_n]$ and then learn regressors $V$ in the second stage. That is, $Z$ is obtained by directly solving (1) but without the $L_1$ penalty term:

$$\min_{Z, Z \in \mathbb{R}^{\widehat{L} \times n}} \|P_\Omega(Y^T Y) - P_\Omega(Z^T Z)\|_F^2 \equiv \min_{\substack{M \succeq 0, \\ rank(M) \leq \widehat{L}}} \|P_\Omega(Y^T Y) - P_\Omega(M)\|_F^2, \qquad (4)$$

where $M = Z^T Z$. Next, $V$ is obtained by solving the following problem:

$$\min_{V \in \mathbb{R}^{\widehat{L} \times d}} \|Z - VX\|_F^2 + \lambda \|V\|_F^2 + \mu \|VX\|_1. \qquad (5)$$

Note that the $Z$ matrix obtained using (4) need not be sparse. However, we store and use $VX$ as our embeddings, so that sparsity is still maintained.

*Optimizing* (4)*:* Note that even the simplified problem (4) is an instance of the popular low-rank matrix completion problem and is known to be NP-hard in general. The main challenge arises due to the non-convex rank constraint on $M$. However, using the Singular Value Projection (SVP) method [20], a popular matrix completion method, we can guarantee convergence to a local minima.

SVP is a simple projected gradient descent method where the projection is onto the set of low-rank matrices. That is, the $t$-th step update for SVP is given by:

$$M_{t+1} = P_{\widehat{L}}(M_t + \eta P_\Omega(Y^T Y - M_t)), \qquad (6)$$

where $M_t$ is the $t$-th step iterate, $\eta > 0$ is the step-size, and $P_{\widehat{L}}(M)$ is the projection of $M$ onto the set of rank-$\widehat{L}$ positive semi-definite definite (PSD) matrices. Note that while the set of rank-$\widehat{L}$ PSD matrices is non-convex, we can still project onto this set efficiently using the eigenvalue decomposition of $M$. That is, if $M = U_M \Lambda_M U_M^T$ be the eigenvalue decomposition of $M$. Then,

$$P_{\widehat{L}}(M) = U_M(1:r) \cdot \Lambda_M(1:r) \cdot U_M(1:r)^T,$$

where $r = \min(\widehat{L}, \widehat{L}_M^+)$ and $\widehat{L}_M^+$ is the number of positive eigenvalues of $M$. $\Lambda_M(1:r)$ denotes the top-$r$ eigenvalues of $M$ and $U_M(1:r)$ denotes the corresponding eigenvectors.

While the above update restricts the rank of all intermediate iterates $M_t$ to be at most $\widehat{L}$, computing rank-$\widehat{L}$ eigenvalue decomposition can still be fairly expensive for large $n$. However, by using special structure in the update (6), one can significantly reduce eigenvalue decomposition's computation complexity as well. In general, the eigenvalue decomposition can be computed in time $O(\widehat{L}\zeta)$ where $\zeta$ is the time complexity of computing a matrix-vector product. Now, for SVP update (6), matrix has special structure of $\hat{M} = M_t + \eta P_\Omega(Y^T Y - M_t)$. Hence $\zeta = O(n\widehat{L} + n\bar{n})$ where $\bar{n} = |\Omega|/n^2$ is the average number of neighbors preserved by SLEEC. Hence, the per-iteration time complexity reduces to $O(n\widehat{L}^2 + n\widehat{L}\bar{n})$ which is linear in $n$, assuming $\bar{n}$ is nearly constant.

*Optimizing* (5)*:* (5) contains an $L_1$ term which makes the problem non-smooth. Moreover, as the $L_1$ term involves both $V$ and $X$, we cannot directly apply the standard prox-function based algorithms. Instead, we use the ADMM method to optimize (5). See Sub-routine 4 for the updates and [21] for a detailed derivation of the algorithm.

**Generalization Error Analysis**: Let $\mathcal{P}$ be a fixed (but unknown) distribution over $\mathcal{X} \times \mathcal{Y}$. Let each training point $(\mathbf{x}_i, \mathbf{y}_i) \in \mathcal{D}$ be sampled i.i.d. from $\mathcal{P}$. Then, the goal of our non-linear embedding method (3) is to learn an embedding matrix $A = V^T V$ that preserves nearest neighbors (in terms of label distance/intersection) of any $(\mathbf{x}, \mathbf{y}) \sim \mathcal{P}$. The above requirements can be formulated as the following stochastic optimization problem:

$$\min_{\substack{A \succeq 0 \\ rank(A) \leq k}} \mathcal{L}(A) = \mathbb{E}_{(\mathbf{x}, \mathbf{y}), (\widetilde{\mathbf{x}}, \widetilde{\mathbf{y}}) \sim \mathcal{P}} \ell(A; (\mathbf{x}, \mathbf{y}), (\widetilde{\mathbf{x}}, \widetilde{\mathbf{y}})), \qquad (7)$$

where the loss function $\ell(A; (\mathbf{x}, \mathbf{y}), (\widetilde{\mathbf{x}}, \widetilde{\mathbf{y}})) = g(\langle \widetilde{\mathbf{y}}, \mathbf{y} \rangle)(\langle \widetilde{\mathbf{y}}, \mathbf{y} \rangle - \widetilde{\mathbf{x}}^T A \mathbf{x})^2$, and $g(\langle \widetilde{\mathbf{y}}, \mathbf{y} \rangle) = \mathbb{I}[\langle \widetilde{\mathbf{y}}, \mathbf{y} \rangle \geq \tau]$, where $\mathbb{I}[\cdot]$ is the indicator function. Hence, a loss is incurred only if $\mathbf{y}$ and $\tilde{\mathbf{y}}$ have a large inner product. For an appropriate selection of the neighborhood selection operator $\Omega$, (3) indeed minimizes a regularized empirical estimate of the loss function (7), i.e., it is a regularized ERM w.r.t. (7).

We now show that the optimal solution $\widehat{A}$ to (3) indeed minimizes the loss (7) upto an additive approximation error. The existing techniques for analyzing excess risk in stochastic optimization require the empirical loss function to be decomposable over the training set, and as such do not apply to (3) which contains loss-terms with two training points. Still, using techniques from the AUC maximization literature [22], we can provide interesting excess risk bounds for Problem (7).

**Theorem 1.** *With probability at least $1 - \delta$ over the sampling of the dataset $\mathcal{D}$, the solution $\widehat{A}$ to the optimization problem* (3) *satisfies*

$$\mathcal{L}(\hat{A}) \le \inf_{A^* \in \mathcal{A}} \left\{ \mathcal{L}(A^*) + \overbrace{C\left(\bar{L}^2 + \left(r^2 + \|A^*\|_F^2\right) R^4\right) \sqrt{\frac{1}{n} \log \frac{1}{\delta}}}^{\text{E-Risk}(n)} \right\},$$

*where $\hat{A}$ is the minimizer of* (3), $r = \frac{\bar{L}}{\lambda}$ *and* $\mathcal{A} := \left\{ A \in \mathbb{R}^{d \times d} : A \succeq 0, rank(A) \le \widehat{L} \right\}$.

See Appendix A for a proof of the result. Note that the generalization error bound is *independent* of both $d$ and $L$, which is critical for extreme multi-label classification problems with large $d, L$. In fact, the error bound is only dependent on $\bar{L} \ll L$, which is the average number of positive labels per data point. Moreover, our bound also provides a way to compute best regularization parameter $\lambda$ that minimizes the error bound. However, in practice, we set $\lambda$ to be a fixed constant.

Theorem 1 only preserves the *population* neighbors of a test point. Theorem 7, given in Appendix A, extends Theorem 1 to ensure that the neighbors in the *training* set are also preserved. We would also like to stress that our excess risk bound is universal and hence holds even if $\hat{A}$ does not minimize (3), i.e., $\mathcal{L}(\hat{A}) \le \mathcal{L}(A^*) + \text{E-Risk}(n) + (\mathcal{L}(\hat{A}) - \mathcal{L}((\hat{A}^*)))$, where E-Risk$(n)$ is given in Theorem 1.

## 2.2 Scaling to Large-scale Data sets

For large-scale data sets, one might require the embedding dimension $\widehat{L}$ to be fairly large (say a few hundreds) which might make computing the updates (6) infeasible. Hence, to scale to such large data sets, SLEEC clusters the given datapoints into smaller local region. Several text-based data sets indeed reveal that there exist small local regions in the feature-space where the number of points as well as the number of labels is reasonably small. Hence, we can train our embedding method over such local regions without significantly sacrificing overall accuracy.

We would like to stress that despite clustering datapoints in homogeneous regions, the label matrix of any given cluster is still not close to low-rank. Hence, applying a state-of-the-art linear embedding method, such as LEML, to each cluster is still significantly less accurate when compared to our method (see Figure 1). Naturally, one can cluster the data set into an extremely large number of regions, so that eventually the label matrix is low-rank in each cluster. However, increasing the number of clusters beyond a certain limit might decrease accuracy as the error incurred during the cluster assignment phase itself might nullify the gain in accuracy due to better embeddings. Figure 1 illustrates this phenomenon where increasing the number of clusters beyond a certain limit in fact decreases accuracy of LEML.

Algorithm 1 provides a pseudo-code of our training algorithm. We first cluster the datapoints into $C$ partitions. Then, for each partition we learn a set of embeddings using Sub-routine 3 and then compute the regression parameters $V^\tau, 1 \le \tau \le C$ using Sub-routine 4. For a given test point $\mathbf{x}$, we first find out the appropriate cluster $\tau$. Then, we find the embedding $\mathbf{z} = V^\tau \mathbf{x}$. The label vector is then predicted using $k$-NN in the embedding space. See Algorithm 2 for more details.

Owing to the curse-of-dimensionality, clustering turns out to be quite unstable for data sets with large $d$ and in many cases leads to some drop in prediction accuracy. To safeguard against such instability, we use an ensemble of models generated using different sets of clusters. We use different initialization points in our clustering procedure to obtain different sets of clusters. Our empirical results demonstrate that using such ensembles leads to significant increase in accuracy of SLEEC (see Figure 2) and also leads to stable solutions with small variance (see Table 4).

## 3 Experiments

Experiments were carried out on some of the largest XML benchmark data sets demonstrating that SLEEC could achieve significantly higher prediction accuracies as compared to the state-of-the-art. It is also demonstrated that SLEEC could be faster at training and prediction than leading embedding techniques such as LEML.

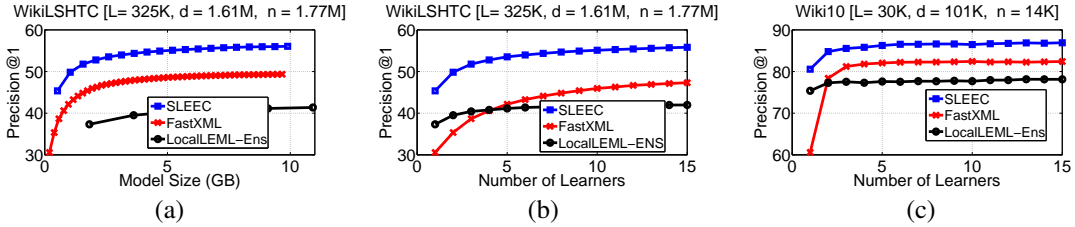

Figure 2: Variation in Precision@1 accuracy with model size and the number of learners on large-scale data sets. Clearly, SLEEC achieves better accuracy than FastXML and LocalLEML-Ensemble at every point of the curve. For WikiLSTHC, SLEEC with a single learner is more accurate than LocalLEML-Ensemble with even 15 learners. Similarly, SLEEC with 2 learners achieves more accuracy than FastXML with 50 learners.

**Data sets**: Experiments were carried out on multi-label data sets including Ads1M [15] (1M labels), Amazon [23] (670K labels), WikiLSHTC (320K labels), DeliciousLarge [24] (200K labels) and Wiki10 [25] (30K labels). All the data sets are publically available except Ads1M which is proprietary and is included here to test the scaling capabilities of SLEEC.

Unfortunately, most of the existing embedding techniques do not scale to such large data sets. We therefore also present comparisons on publically available small data sets such as BibTeX [26], MediaMill [27], Delicious [28] and EURLex [29]. (Table 2 in the appendix lists their statistics).

**Baseline algorithms**: This paper's primary focus is on comparing SLEEC to state-of-the-art methods which can scale to the large data sets such as embedding based LEML [9] and tree based FastXML [15] and LPSR [2]. Naïve Bayes was used as the base classifier in LPSR as was done in [15]. Techniques such as CS [3], CPLST [30], ML-CSSP [7], 1-vs-All [31] could only be trained on the small data sets given standard resources. Comparisons between SLEEC and such techniques are therefore presented in the supplementary material. The implementation for LEML and FastXML was provided by the authors. We implemented the remaining algorithms and ensured that the published results could be reproduced and were verified by the authors wherever possible.

**Hyper-parameters**: Most of SLEEC's hyper-parameters were kept fixed including the number of clusters in a learner $\left(\lfloor N_{\text{Train}}/6000\rfloor\right)$, embedding dimension (100 for the small data sets and 50 for the large), number of learners in the ensemble (15), and the parameters used for optimizing (3). The remaining two hyper-parameters, the $k$ in kNN and the number of neighbours considered during SVP, were both set by limited validation on a validation set.

The hyper-parameters for all the other algorithms were set using fine grained validation on each data set so as to achieve the highest possible prediction accuracy for each method. In addition, all the embedding methods were allowed a much larger embedding dimension $(0.8L)$ than SLEEC (100) to give them as much opportunity as possible to outperform SLEEC.

**Evaluation Metrics**: We evaluated algorithms using metrics that have been widely adopted for XML and ranking tasks. Precision at $k$ (P@$k$) is one such metric that counts the fraction of correct predictions in the top $k$ scoring labels in $\hat{\mathbf{y}}$, and has been widely utilized [1, 3, 15, 13, 2, 9]. We use the ranking measure nDCG@$k$ as another evaluation metric. We refer the reader to the supplementary material – Appendix B.1 and Tables 5 and 6 – for further descriptions of the metrics and results.

**Results on large data sets with more than 100K labels**: Table 1a compares SLEEC's prediction accuracy, in terms of P@$k$ (k= $\{1, 3, 5\}$), to all the leading methods that could be trained on five such data sets. SLEEC could improve over the leading embedding method, LEML, by as much as 35% and 15% in terms of P@1 and P@5 on WikiLSHTC. Similarly, SLEEC outperformed LEML by 27% and 22% in terms of P@1 and P@5 on the Amazon data set which also has many tail labels. The gains on the other data sets are consistent, but smaller, as the tail label problem is not so acute. SLEEC also outperforms the leading tree method, FastXML, by 6% in terms of both P@1 and P@5 on WikiLSHTC and Wiki10 respectively. This demonstrates the superiority of SLEEC's overall pipeline constructed using local distance preserving embeddings followed by kNN classification.

SLEEC also has better scaling properties as compared to all other embedding methods. In particular, apart from LEML, no other embedding approach could scale to the large data sets and, even LEML could not scale to Ads1M with a million labels. In contrast, a single SLEEC learner could be learnt on WikiLSHTC in 4 hours on a single core and already gave ∼ 20% improvement in P@1 over LEML (see Figure 2 for the variation in accuracy vs SLEEC learners). In fact, SLEEC's training

Table 1: **Precision Accuracies** (a) Large-scale data sets : Our proposed method SLEEC is as much as 35% more accurate in terms of P@1 and 22% in terms of P@5 than LEML, a leading embedding method. Other embedding based methods do not scale to the large-scale data sets; we compare against them on small-scale data sets in Table 3. SLEEC is also 6% more accurate (w.r.t. P@1 and P@5) than FastXML, a state-of-the-art tree method. '-' indicates LEML could not be run with the standard resources. (b) Small-scale data sets : SLEEC consistently outperforms state of the art approaches. WSABIE, which also uses kNN classifier on its embeddings is significantly less accurate than SLEEC on all the data sets, showing the superiority of our embedding learning algorithm.

(a)

| Data set | | SLEEC | LEML | FastXML | LPSR-NB |
|---|---|---|---|---|---|
| Wiki10 | P@1 | **85.54** | 73.50 | 82.56 | 72.71 |
| | P@3 | **73.59** | 62.38 | 66.67 | 58.51 |
| | P@5 | **63.10** | 54.30 | 56.70 | 49.40 |
| Delicious-Large | P@1 | **47.03** | 40.30 | 42.81 | 18.59 |
| | P@3 | **41.67** | 37.76 | 38.76 | 15.43 |
| | P@5 | **38.88** | 36.66 | 36.34 | 14.07 |
| WikiLSHTC | P@1 | **55.57** | 19.82 | 49.35 | 27.43 |
| | P@3 | **33.84** | 11.43 | 32.69 | 16.38 |
| | P@5 | **24.07** | 8.39 | 24.03 | 12.01 |
| Amazon | P@1 | **35.05** | 8.13 | 33.36 | 28.65 |
| | P@3 | **31.25** | 6.83 | 29.30 | 24.88 |
| | P@5 | **28.56** | 6.03 | 26.12 | 22.37 |
| Ads-1m | P@1 | 21.84 | - | **23.11** | 17.08 |
| | P@3 | **14.30** | - | 13.86 | 11.38 |
| | P@5 | **11.01** | - | 10.12 | 8.83 |

(b)

| Data set | | SLEEC | LEML | FastXML | WSABIE | OneVsAll |
|---|---|---|---|---|---|---|
| BibTex | P@1 | **65.57** | 62.53 | 63.73 | 54.77 | 61.83 |
| | P@3 | **40.02** | 38.4 | 39.00 | 32.38 | 36.44 |
| | P@5 | **29.30** | 28.21 | 28.54 | 23.98 | 26.46 |
| Delicious | P@1 | 68.42 | 65.66 | **69.44** | 64.12 | 65.01 |
| | P@3 | 61.83 | 60.54 | **63.62** | 58.13 | 58.90 |
| | P@5 | 56.80 | 56.08 | **59.10** | 53.64 | 53.26 |
| MediaMill | P@1 | **87.09** | 84.00 | 84.24 | 81.29 | 83.57 |
| | P@3 | **72.44** | 67.19 | 67.39 | 64.74 | 65.50 |
| | P@5 | **58.45** | 52.80 | 53.14 | 49.82 | 48.57 |
| EurLEX | P@1 | **80.17** | 61.28 | 68.69 | 70.87 | 74.96 |
| | P@3 | **65.39** | 48.66 | 57.73 | 56.62 | 62.92 |
| | P@5 | **53.75** | 39.91 | 48.00 | 46.2 | 53.42 |

time on WikiLSHTC was comparable to that of tree based FastXML. FastXML trains 50 trees in 7 hours on a single core to achieve a P@1 of 49.37% whereas SLEEC could achieve 49.98% by training 2 learners in 8 hours. Similarly, SLEEC's training time on Ads1M was 6 hours per learner on a single core.

SLEEC's predictions could also be up to 300 times faster than LEMLs. For instance, on WikiLSHTC, SLEEC made predictions in 8 milliseconds per test point as compared to LEML's 279. SLEEC therefore brings the prediction time of embedding methods to be much closer to that of tree based methods (FastXML took 0.5 milliseconds per test point on WikiLSHTC) and within the acceptable limit of most real world applications.

**Effect of clustering and multiple learners**: As mentioned in the introduction, other embedding methods could also be extended by clustering the data and then learning a local embedding in each cluster. Ensembles could also be learnt from multiple such clusterings. We extend LEML in such a fashion, and refer to it as LocalLEML, by using exactly the same 300 clusters per learner in the ensemble as used in SLEEC for a fair comparison. As can be seen in Figure 2, SLEEC significantly outperforms LocalLEML with a single SLEEC learner being much more accurate than an ensemble of even 10 LocalLEML learners. Figure 2 also demonstrates that SLEEC's ensemble can be much more accurate at prediction as compared to the tree based FastXML ensemble (the same plot is also presented in the appendix depicting the variation in accuracy with model size in RAM rather than the number of learners in the ensemble). The figure also demonstrates that very few SLEEC learners need to be trained before accuracy starts saturating. Finally, Table 4 shows that the variance in SLEEC s prediction accuracy (w.r.t. different cluster initializations) is very small, indicating that the method is stable even though clustering in more than a million dimensions.

**Results on small data sets**: Table 3, in the appendix, compares the performance of SLEEC to several popular methods including embeddings, trees, kNN and 1-vs-All SVMs. Even though the tail label problem is not acute on these data sets, and SLEEC was restricted to a single learner, SLEEC's predictions could be significantly more accurate than all the other methods (except on Delicious where SLEEC was ranked second). For instance, SLEEC could outperform the closest competitor on EurLex by 3% in terms of P1. Particularly noteworthy is the observation that SLEEC outperformed WSABIE [13], which performs kNN classification on linear embeddings, by as much as 10% on multiple data sets. This demonstrates the superiority of SLEEC's local distance preserving embeddings over the traditional low-rank embeddings.

## Acknowledgments

We are grateful to Abhishek Kadian for helping with the experiments. Himanshu Jain is supported by a Google India PhD Fellowship at IIT Delhi

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
