[Supplementary Material]

# A Generalization Error Analysis

To present our results, we first introduce some notation: for any embedding matrix $A$ and dataset $\mathcal{D}$, let

$$\hat{\mathcal{L}}(A; \mathcal{D}) := \frac{1}{n(n-1)} \sum_{i=1}^{n} \sum_{j \neq i} \ell(A; (\mathbf{x}_i, \mathbf{y}_i), (\mathbf{x}_j, \mathbf{y}_j))$$

$$\tilde{\mathcal{L}}(A; \mathcal{D}) := \frac{1}{n} \sum_{i=1}^{n} \mathop{\mathbb{E}}_{(\mathbf{x}, \mathbf{y}) \sim \mathcal{P}} \ell(A; (\mathbf{x}, \mathbf{y}), (\mathbf{x}_i, \mathbf{y}_i))$$

$$\mathcal{L}(A) := \mathop{\mathbb{E}}_{(\mathbf{x}, \mathbf{y}),(\tilde{\mathbf{x}}, \tilde{\mathbf{y}}) \sim \mathcal{P}} \ell(A; (\mathbf{x}, \mathbf{y}), (\tilde{\mathbf{x}}, \tilde{\mathbf{y}}))$$

We assume, without loss of generality that the data points are confined to a unit ball i.e. $\|\mathbf{x}\|_2 \leq 1$ for all $\mathbf{x} \in \mathcal{X}$. Also let $Q = C \cdot (\bar{L}(r + \bar{L}))$ where $\bar{L}$ is the average number of labels active in a data point, $r = \frac{\bar{L}}{\lambda}$, $\lambda$ and $\mu$ are the regularization constants used in (3), and $C$ is a *universal constant*.

**Theorem 1.** *Assume that all data points are confined to a ball of radius $R$ i.e $\|x\|_2 \leq R$ for all $x \in \mathcal{X}$. Then with probability at least $1 - \delta$ over the sampling of the data set $\mathcal{D}$, the solution $\hat{A}$ to the optimization problem* (3) *satisfies,*

$$\mathcal{L}(\hat{A}) \leq \inf_{A^* \in \mathcal{A}} 8 \left\{ \mathcal{L}(A^*) + C \left( \bar{L}^2 + \left(r^2 + \|A^*\|_F^2\right) R^4 \right) \sqrt{\frac{1}{n} \log \frac{1}{\delta}} \right\},$$

*where $r = \frac{\bar{L}}{\lambda}$, and $C$ and $C'$ are universal constants.*

*Proof.* Our proof will proceed in the following steps. Let $A^*$ be the population minimizer of the objective in the statement of the theorem.

1. **Step 1** (Capacity bound): we will show that for some $r$, we have $\|\hat{A}\|_F \leq r$

2. **Step 2** (Uniform convergence): we will show that w.h.p., $\sup_{\substack{A \in \mathcal{A} \\ \|A\| \leq r}} \left\{ \mathcal{L}(A) - \hat{\mathcal{L}}(A; \mathcal{D}) \right\} \leq \mathcal{O}\left( \sqrt{\frac{1}{n} \log \frac{1}{\delta}} \right)$

3. **Step 3** (Point convergence): we will show that w.h.p., $\hat{\mathcal{L}}(A^*; \mathcal{D}) - \mathcal{L}(A^*) \leq \mathcal{O}\left( \sqrt{\frac{1}{n} \log \frac{1}{\delta}} \right)$

Having these results will allow us to prove the theorem in the following manner

$$\mathcal{L}(\hat{A}) \leq \hat{\mathcal{L}}(\hat{A}, \mathcal{D}) + \sup_{\substack{A \in \mathcal{A} \\ \|A\| \leq r}} \left\{ \hat{\mathcal{L}}(A; \mathcal{D}) - \mathcal{L}(A) \right\} \leq \hat{\mathcal{L}}(A^*, \mathcal{D}) + \mathcal{O}\left( \sqrt{\frac{1}{n} \log \frac{1}{\delta}} \right) \leq \mathcal{L}(A^*) + \mathcal{O}\left( \sqrt{\frac{1}{n} \log \frac{1}{\delta}} \right),$$

where the second step follows from the fact that $\hat{A}$ is the empirical risk minimizer.

We will now prove these individual steps as separate lemmata, where we will also reveal the exact constants in these results.

**Lemma 2** (Capacity bound)**.** *For the regularization parameters chosen for the loss function $\ell(\cdot)$, the following holds for the minimizer $\hat{A}$ of* (3)

$$\|\hat{A}\|_F \leq Tr(A) \leq \frac{1}{\lambda} \bar{L}.$$

*Proof.* Since, $\hat{A}$ minimizes (3), we have:

$$\|A\|_F \leq Tr(A) \leq \frac{1}{\lambda} \frac{1}{n(n-1)} \sum_{ij} (\langle \mathbf{y}_i, \mathbf{y}_j \rangle)^2 \leq \frac{1}{\lambda} \max_{ij} \langle \mathbf{y}_i, \mathbf{y}_j \rangle.$$

$\square$

The above result shows that we can, for future analysis, restrict our hypothesis space to

$$\widetilde{\mathcal{A}}(r) := \left\{ A \in \mathcal{A} : \|A\|_F^2 \leq r^2 \right\},$$

where we set $r = \frac{\bar{L}}{\lambda}$. This will be used to prove the following result.

**Lemma 3** (Uniform convergence)**.** *With probability at least $1 - \delta$ over the choice of the data set $\mathcal{D}$, we have*

$$\hat{\mathcal{L}}(\hat{A}; \mathcal{D}) - \mathcal{L}(\hat{A}) \leq 6 \left( rR^2 + \bar{L} \right)^2 \sqrt{\frac{1}{2n} \log \frac{1}{\delta}}$$

*Proof.* For notional simplicity, we will denote a labeled sample as $\mathbf{z} = (\mathbf{x}, \mathbf{y})$. Given any two points $\mathbf{z}, \mathbf{z}' \in \mathcal{Z} = \mathcal{X} \times \mathcal{Y}$ and any $A \in \widetilde{\mathcal{A}}(r)$, we will then write

$$\ell(A; \mathbf{z}, \mathbf{z}') = g(\langle \mathbf{y}, \mathbf{y}' \rangle) \left( \langle \mathbf{y}, \mathbf{y}' \rangle - \mathbf{x}^T A \mathbf{x}' \right)^2 + \lambda Tr(A),$$

so that, for the training set $\mathcal{D} = \{ \mathbf{z}_1, \ldots, \mathbf{z}_n \}$, we have

$$\hat{\mathcal{L}}(A; \mathcal{D}) = \frac{1}{n(n-1)} \sum_{i=1}^{n} \sum_{j \neq i} \ell(A; \mathbf{z}_i, \mathbf{z}_j)$$

as well as

$$\mathcal{L}(A) = \mathop{\mathbb{E}}_{z, z' \sim \mathcal{P}} \ell(A; \mathbf{z}, \mathbf{z}').$$

Note that we ignore the $\|V\mathbf{x}\|_1$ term in (3) completely because it is a regularization term and it won't increase the excess risk.

Suppose we draw a fresh data set $\widetilde{\mathcal{D}} = \{ \tilde{\mathbf{z}}_1, \ldots, \tilde{\mathbf{z}}_n \} \sim \mathcal{P}$, then we have, by linearity of expectation,

$$\mathop{\mathbb{E}}_{\widetilde{\mathcal{D}} \sim \mathcal{P}} \hat{\mathcal{L}}(\hat{A}; \tilde{\mathcal{D}}) = \frac{1}{n(n-1)} \sum_{i=1}^{n} \sum_{j \neq i} \mathop{\mathbb{E}}_{\widetilde{\mathcal{D}} \sim \mathcal{P}} \ell(A; \tilde{\mathbf{z}}_i, \tilde{\mathbf{z}}_j) = \mathcal{L}(\hat{A}).$$

Now notice that for any $A \in \widetilde{\mathcal{A}}$, suppose we perturb the data set $\mathcal{D}$ at the $i$th location to get a perturbed data set $\mathcal{D}^i$, then the following holds

$$\left| \hat{\mathcal{L}}(A; \mathcal{D}) - \hat{\mathcal{L}}(A; \mathcal{D}^i) \right| \leq \frac{4(\bar{L}^2 + r^2 R^4)}{n}.$$

which allows us to bound the excess risk as follows

$$\mathcal{L}(\hat{A}) - \hat{\mathcal{L}}(\hat{A}; \mathcal{D}) = \mathop{\mathbb{E}}_{\widetilde{\mathcal{D}} \sim \mathcal{P}} \hat{\mathcal{L}}(\hat{A}; \tilde{\mathcal{D}}) - \hat{\mathcal{L}}(\hat{A}; \mathcal{D}) \leq \sup_{A \in \widetilde{\mathcal{A}}(r)} \left\{ \mathop{\mathbb{E}}_{\widetilde{\mathcal{D}} \sim \mathcal{P}} \hat{\mathcal{L}}(A; \tilde{\mathcal{D}}) - \hat{\mathcal{L}}(A; \mathcal{D}) \right\}$$

$$\leq \mathop{\mathbb{E}}_{\mathcal{D} \sim \mathcal{P}} \sup_{A \in \widetilde{\mathcal{A}}(r)} \left\{ \mathop{\mathbb{E}}_{\widetilde{\mathcal{D}} \sim \mathcal{P}} \hat{\mathcal{L}}(A; \tilde{\mathcal{D}}) - \hat{\mathcal{L}}(A; \mathcal{D}) \right\} + 4(\bar{L}^2 + r^2 R^4) \sqrt{\frac{1}{2n} \log \frac{1}{\delta}}$$

$$\leq \underbrace{\mathop{\mathbb{E}}_{\mathcal{D}, \widetilde{\mathcal{D}} \sim \mathcal{P}} \sup_{A \in \widetilde{\mathcal{A}}(r)} \left\{ \hat{\mathcal{L}}(A; \tilde{\mathcal{D}}) - \hat{\mathcal{L}}(A; \mathcal{D}) \right\}}_{Q_n(\widetilde{\mathcal{A}}(r))} + 4(\bar{L}^2 + r^2 R^4) \sqrt{\frac{1}{2n} \log \frac{1}{\delta}},$$

where the third step follows from an application of McDiarmid's inequality and the last step follows from Jensen's inequality. We now bound the quantity $Q_n(\widetilde{\mathcal{A}}(r))$ below. Let $\bar{\ell}(A, \mathbf{z}, \mathbf{z}') :=$

$\ell(A, \mathbf{z}, \mathbf{z}') - \lambda \cdot Tr(A)$. Then we have

$$
Q_n(\widetilde{\mathcal{A}}(r)) = \mathop{\mathbb{E}}_{\mathcal{D}, \tilde{\mathcal{D}} \sim \mathcal{P}} \sup_{A \in \widetilde{\mathcal{A}}(r)} \left\{ \hat{\mathcal{L}}(A; \tilde{\mathcal{D}}) - \hat{\mathcal{L}}(A; \mathcal{D}) \right\}
$$

$$
= \frac{1}{n(n-1)} \mathop{\mathbb{E}}_{\mathbf{z}_i, \tilde{\mathbf{z}}_i \sim \mathcal{P}} \left[\!\!\left[ \sup_{A \in \widetilde{\mathcal{A}}(r)} \left\{ \sum_{i=1}^{n} \sum_{j \neq i} \ell(A; \tilde{\mathbf{z}}_i, \tilde{\mathbf{z}}_j) - \ell(A; \mathbf{z}_i, \mathbf{z}_j) \right\} \right]\!\!\right]
$$

$$
= \frac{1}{n(n-1)} \mathop{\mathbb{E}}_{\mathbf{z}_i, \tilde{\mathbf{z}}_i \sim \mathcal{P}} \left[\!\!\left[ \sup_{A \in \widetilde{\mathcal{A}}(r)} \left\{ \sum_{i=1}^{n} \sum_{j \neq i} \bar{\ell}(A; \tilde{\mathbf{z}}_i, \tilde{\mathbf{z}}_j) - \bar{\ell}(A; \mathbf{z}_i, \mathbf{z}_j) \right\} \right]\!\!\right]
$$

$$
\leq \frac{2}{n} \mathop{\mathbb{E}}_{\mathbf{z}_i, \tilde{\mathbf{z}}_i} \left[\!\!\left[ \sup_{A \in \widetilde{\mathcal{A}}(r)} \left\{ \sum_{i=1}^{n/2} \bar{\ell}(A; \tilde{\mathbf{z}}_i, \tilde{\mathbf{z}}_{n/2+i}) - \bar{\ell}(A; \mathbf{z}_i, \mathbf{z}_{n/2+i}) \right\} \right]\!\!\right]
$$

$$
\leq 2 \cdot \underbrace{\frac{2}{n} \mathop{\mathbb{E}}_{z_i, \epsilon_i} \left[\!\!\left[ \sup_{A \in \widetilde{\mathcal{A}}(r)} \left\{ \sum_{i=1}^{n/2} \epsilon_i \bar{\ell}(A; \mathbf{z}_i, \mathbf{z}_{n/2+i}) \right\} \right]\!\!\right]}_{\mathcal{R}_n(\ell \circ \widetilde{\mathcal{A}}(r))} = 2 \cdot \mathcal{R}_{n/2}(\ell \circ \widetilde{\mathcal{A}}(r))
$$

where the last step uses a standard symmetrization argument with the introduction of the Rademacher variables $\epsilon_i \sim -1, +1$. The second step presents a stumbling block in the analysis since the interaction between the pairs of the points means that traditional symmetrization can no longer done. Previous works analyzing such "pairwise" loss functions face similar problems [22]. Consequently, this step uses a powerful alternate representation for U-statistics to simplify the expression. This technique is attributed to Serfling. This, along with the Hoeffding decomposition, are two of the most powerful techniques to deal with "coupled" random variables as we have in this situation.

**Theorem 4.** *For any set of real valued functions $q_\tau : \mathcal{X} \times \mathcal{X} \to \mathbb{R}$ indexed by $\tau \in T$, if $X_1, \ldots, X_n$ are i.i.d. random variables then we have*

$$
\mathbb{E} \left[\!\!\left[ \sup_{\tau \in T} \frac{2}{n(n-1)} \sum_{1 \leq i < j \leq n} q_\tau(X_i, X_j) \right]\!\!\right] \leq \mathbb{E} \left[\!\!\left[ \sup_{\tau \in T} \frac{2}{n} \sum_{i=1}^{n/2} q_\tau(X_i, X_{n/2+i}) \right]\!\!\right]
$$

Applying this decoupling result to the random variables $X_i = (\tilde{\mathbf{z}}_i, \mathbf{z}_i)$, the index set $\widetilde{\mathcal{A}}(r)$ and functions $q_A(X_i, X_j) = \ell(A; \tilde{\mathbf{z}}_i, \tilde{\mathbf{z}}_j) - \ell(A; \mathbf{z}_i, \mathbf{z}_j) = \bar{\ell}(A; \tilde{\mathbf{z}}_i, \tilde{\mathbf{z}}_j) - \bar{\ell}(A; \mathbf{z}_i, \mathbf{z}_j)$ gives us the second step. We now concentrate on bounding the resulting Rademacher average term $\mathcal{R}_n(\ell \circ \widetilde{\mathcal{A}}(r))$. We have

$$
\mathcal{R}_{n/2}(\ell \circ \widetilde{\mathcal{A}}(r)) = \frac{2}{n} \mathop{\mathbb{E}}_{z_i, \epsilon_i} \left[\!\!\left[ \sup_{A \in \widetilde{\mathcal{A}}(r)} \left\{ \sum_{i=1}^{n/2} \epsilon_i \bar{\ell}(A; \mathbf{z}_i, \mathbf{z}_{n/2+i}) \right\} \right]\!\!\right]
$$

$$
= \frac{2}{n} \mathop{\mathbb{E}}_{z_i, \epsilon_i} \left[\!\!\left[ \sup_{A \in \widetilde{\mathcal{A}}(r)} \left\{ \sum_{i=1}^{n/2} \epsilon_i g(\langle \mathbf{y}_i, \mathbf{y}_{n/2+i} \rangle) \left( \langle \mathbf{y}_i, \mathbf{y}_{n/2+i} \rangle - \mathbf{x}_i^T A \mathbf{x}_{n/2+i} \right)^2 \right\} \right]\!\!\right] .
$$

That is,

$$\mathcal{R}_{n/2}(\ell \circ \widetilde{\mathcal{A}}(r)) \leq \underbrace{\frac{2}{n} \mathop{\mathbb{E}}_{z_i,\epsilon_i} \left[\!\!\left[ \sum_{i=1}^{n/2} \epsilon_i g(\langle \mathbf{y}_i, \mathbf{y}_{n/2+i}\rangle) \langle \mathbf{y}_i, \mathbf{y}_{n/2+i}\rangle^2 \right]\!\!\right]}_{(A)}$$

$$+ \underbrace{\frac{2}{n} \mathop{\mathbb{E}}_{z_i,\epsilon_i} \left[\!\!\left[ \sup_{A \in \widetilde{\mathcal{A}}(r)} \left\{ \sum_{i=1}^{n/2} \epsilon_i g(\langle \mathbf{y}_i, \mathbf{y}_{n/2+i}\rangle) \left(\mathbf{x}_i^T A \mathbf{x}_{n/2+i}\right)^2 \right\} \right]\!\!\right]}_{B_n(\ell \circ \widetilde{\mathcal{A}}(r))}$$

$$+ \underbrace{\frac{4}{n} \mathop{\mathbb{E}}_{z_i,\epsilon_i} \left[\!\!\left[ \sup_{A \in \widetilde{\mathcal{A}}(r)} \left\{ \sum_{i=1}^{n/2} \epsilon_i g(\langle \mathbf{y}_i, \mathbf{y}_{n/2+i}\rangle) \langle \mathbf{y}_i, \mathbf{y}_{n/2+i}\rangle \left(\mathbf{x}_i^T A \mathbf{x}_{n/2+i}\right) \right\} \right]\!\!\right]}_{C_n(\ell \circ \widetilde{\mathcal{A}}(r))}$$

Now since the random variables $\epsilon_i$ are zero mean and independent of $z_i$, we have $\mathop{\mathbb{E}}_{\epsilon_i | z_i, z_{n/2+i}} \epsilon_i = 0$ which we can use to show that $\mathop{\mathbb{E}}_{\epsilon_i | z_i, z_{n/2+i}} \left[\!\!\left[ \epsilon_i g(\langle \mathbf{y}_i, \mathbf{y}_{n/2+i}\rangle) \langle \mathbf{y}_i, \mathbf{y}_{n/2+i}\rangle^2 \right]\!\!\right] = 0$ which gives us, by linearity of expectation, $(A) = 0$. To bound the next two terms we use the following standard contraction inequality:

**Theorem 5.** *Let $\mathcal{H}$ be a set of bounded real valued functions from some domain $\mathcal{X}$ and let $\mathbf{x}_1, \ldots, \mathbf{x}_n$ be arbitrary elements from $\mathcal{X}$. Furthermore, let $\phi_i : \mathbb{R} \to \mathbb{R}$, $i = 1, \ldots, n$ be $L$-Lipschitz functions such that $\phi_i(0) = 0$ for all $i$. Then we have*

$$\mathbb{E}\left[\!\!\left[ \sup_{h \in \mathcal{H}} \frac{1}{n} \sum_{i=1}^{n} \epsilon_i \phi_i(h(\mathbf{x}_i)) \right]\!\!\right] \leq L\mathbb{E}\left[\!\!\left[ \sup_{h \in \mathcal{H}} \frac{1}{n} \sum_{i=1}^{n} \epsilon_i h(\mathbf{x}_i) \right]\!\!\right].$$

Now define

$$\phi_i(w) = g(\langle \mathbf{y}_i, \mathbf{y}_{n/2+i}\rangle) w^2$$

Clearly $\phi_i(0) = 0$ and $0 \leq g(\langle \mathbf{y}_i, \mathbf{y}_{n/2+i}\rangle) \leq 1$. Moreover, in our case $w = x^T A x'$ for some $A \in \widetilde{\mathcal{A}}(r)$ and $\|x\|, \|x'\| \leq R$. Thus, the function $\phi_i(\cdot)$ is $rR^2$-Lipschitz. Note that here we exploit the fact that the contraction inequality is actually proven for the empirical Rademacher averages due to which we can take $g(\langle \mathbf{y}_i, \mathbf{y}_{n/2+i}\rangle)$ to be a constant dependent only on $i$. This allows us to bound the term $B_n(\ell \circ \widetilde{\mathcal{A}}(r))$ as follows

$$B_n(\ell \circ \widetilde{\mathcal{A}}(r)) = \frac{2}{n} \mathop{\mathbb{E}}_{z_i,\epsilon_i} \left[\!\!\left[ \sup_{A \in \widetilde{\mathcal{A}}(r)} \left\{ \sum_{i=1}^{n/2} \epsilon_i g(\langle \mathbf{y}_i, \mathbf{y}_{n/2+i}\rangle) \left(\mathbf{x}_i^T A \mathbf{x}_{n/2+i}\right)^2 \right\} \right]\!\!\right]$$

$$\leq rR^2 \cdot \underbrace{\frac{2}{n} \mathop{\mathbb{E}}_{z_i,\epsilon_i} \left[\!\!\left[ \sup_{A \in \widetilde{\mathcal{A}}(r)} \left\{ \sum_{i=1}^{n/2} \epsilon_i \left(\mathbf{x}_i^T A \mathbf{x}_{n/2+i}\right) \right\} \right]\!\!\right]}_{\mathcal{R}_{n/2}(\widetilde{\mathcal{A}}(r))} \leq rR^2 \cdot \mathcal{R}_{n/2}(\widetilde{\mathcal{A}}(r)).$$

Similarly, we can show that

$$C_n(\ell \circ \widetilde{\mathcal{A}}(r)) = \frac{4}{n} \mathop{\mathbb{E}}_{z_i,\epsilon_i} \left[\!\!\left[ \sup_{A \in \widetilde{\mathcal{A}}(r)} \left\{ \sum_{i=1}^{n/2} \epsilon_i g(\langle \mathbf{y}_i, \mathbf{y}_{n/2+i}\rangle) \langle \mathbf{y}_i, \mathbf{y}_{n/2+i}\rangle \left(\mathbf{x}_i^T A \mathbf{x}_{n/2+i}\right) \right\} \right]\!\!\right]$$

$$\leq \frac{4\bar{L}}{n} \mathop{\mathbb{E}}_{z_i,\epsilon_i} \left[\!\!\left[ \sup_{A \in \widetilde{\mathcal{A}}(r)} \left\{ \sum_{i=1}^{n/2} \epsilon_i \left(\mathbf{x}_i^T A \mathbf{x}_{n/2+i}\right) \right\} \right]\!\!\right]$$

$$\leq 2\bar{L} \cdot \mathcal{R}_{n/2}(\widetilde{\mathcal{A}}(r)).$$

Thus, we have

$$\mathcal{R}_{n/2}(\ell \circ \widetilde{\mathcal{A}}(r)) \leq \left(rR^2 + 2\bar{L}\right) \cdot \mathcal{R}_{n/2}(\widetilde{\mathcal{A}}(r))$$

Now all that remains to be done is bound $\mathcal{R}_n(\ell \circ \widetilde{\mathcal{A}}(r))$. This can be done by invoking standard bounds on Rademacher averages for regularized function classes. In particular, using the two stage proof technique outlined in [22], we can show that

$$\mathcal{R}_{n/2}(\widetilde{\mathcal{A}}(r)) \leq rR^2 \sqrt{\frac{2}{n}}$$

Putting it all together gives us the following bound: with probability at least $1 - \delta$, we have

$$\mathcal{L}(\hat{A}) - \hat{\mathcal{L}}(\hat{A}; \mathcal{D}) \leq 2(rR^2 + 2\bar{L})rR^2\sqrt{\frac{2}{n}} + 4(\bar{L}^2 + r^2R^4)\sqrt{\frac{1}{2n}\log\frac{1}{\delta}}$$

as claimed $\qquad \square$

The final part shows pointwise convergence for the population risk minimizer.

**Lemma 6** (Point convergence)**.** *With probability at least $1 - \delta$ over the choice of the data set $\mathcal{D}$, we have*

$$\hat{\mathcal{L}}(A^*; \mathcal{D}) - \mathcal{L}(A^*) \leq 4(\bar{L}^2 + \|A^*\|_F^2 R^4)\sqrt{\frac{1}{2n}\log\frac{1}{\delta}},$$

*where $A^*$ is the population minimizer of the objective in the theorem statement.*

*Proof.* We note that, as before

$$\underset{\mathcal{D}\sim\mathcal{P}}{\mathbb{E}}\hat{\mathcal{L}}(A^*, \mathcal{D}) = \mathcal{L}(A^*)$$

Let $\mathcal{D}$ be a realization of the sample and $\mathcal{D}^i$ be a perturbed data set where the $i^{\text{th}}$ data point is arbitrarily perturbed. Then we have

$$\left|\hat{\mathcal{L}}(A^*; \mathcal{D}) - \hat{\mathcal{L}}(A^*; \mathcal{D}^i)\right| \leq \frac{4\left(\bar{L}^2 + \|A^*\|_F^2 R^4\right)}{n}.$$

Thus, an application of McDiarmid's inequality shows us that with probability at least $1 - \delta$, we have

$$\hat{\mathcal{L}}(A^*; \mathcal{D}) - \mathcal{L}(A^*) = \hat{\mathcal{L}}(A^*; \mathcal{D}) - \underset{\mathcal{D}\sim\mathcal{P}}{\mathbb{E}}\hat{\mathcal{L}}(A^*; \mathcal{D}) \leq 4\left(\bar{L}^2 + \|A^*\|_F^2 R^4\right)\sqrt{\frac{1}{2n}\log\frac{1}{\delta}},$$

which proves the claim. $\qquad \square$

Putting the three lemmata together as shown above concludes the proof of the theorem. $\qquad \square$

Although the above result ensures that the embedding provided by $\hat{A}$ would preserve neighbors over the population, in practice, we are more interested in preserving the neighbors of test points among the training points, as they are used to predict the label vector. The following extension of our result shows that $\hat{A}$ indeed accomplishes this as well.

**Theorem 7.** *Assume that all data points are confined to a ball of radius $R$ i.e $\|x\|_2 \leq R$ for all $x \in \mathcal{X}$. Then with probability at least $1 - \delta$ over the sampling of the data set $\mathcal{D}$, the solution $\hat{A}$ to the optimization problem* (3) *ensures that,*

$$\tilde{\mathcal{L}}(\hat{A}; \mathcal{D}) \leq \inf_{A^* \in \mathcal{A}}\left\{\tilde{\mathcal{L}}(A^*; \mathcal{D}) + C\left(\bar{L}^2 + \left(r^2 + \|A^*\|_F^2\right)R^4\right)\sqrt{\frac{1}{n}\log\frac{1}{\delta}}\right\},$$

*where $r = \frac{\bar{L}}{\lambda}$, and $C$ is a universal constant.*

Note that the loss function $\tilde{\mathcal{L}}(A; \mathcal{D})$ exactly captures the notion of how well an embedding matrix $A$ can preserve the neighbors of an unseen point among the training points.

*Proof.* We first recall and rewrite the form of the loss function considered here. For any data set $\mathcal{D} = \{z_1, \ldots, z_n\}$. For any $A \in \mathcal{A}$ and $z \in \mathbb{Z}$, let $\wp(A; z) := \mathop{\mathbb{E}}_{z' \sim \mathcal{P}} \ell(A; z, z')$. This allows us to write

$$\tilde{\mathcal{L}}(A; \mathcal{D}) := \frac{1}{n} \sum_{i=1}^{n} \mathop{\mathbb{E}}_{z \sim \mathcal{P}} \ell(A; z, z_i) = \frac{1}{n} \sum_{i=1}^{n} \wp(A; z_i)$$

Also note that for any fixed $A$, we have

$$\mathop{\mathbb{E}}_{\mathcal{D} \sim \mathcal{P}} \tilde{\mathcal{L}}(A; \mathcal{D}) = \mathcal{L}(A).$$

Now, given a perturbed data set $\mathcal{D}^i$, we have

$$\left| \tilde{\mathcal{L}}(A; \mathcal{D}) - \tilde{\mathcal{L}}(A; \mathcal{D}^i) \right| \leq \frac{4(\bar{L}^2 + r^2 R^4)}{n},$$

as before. Since this problem does not have to take care of pairwise interactions between the data points (since the "other" data point is being taken expectations over), using standard Rademacher style analysis gives us, with probability at least $1 - \delta$,

$$\tilde{\mathcal{L}}(\hat{A}; \mathcal{D}) - \mathcal{L}(\hat{A}) \leq 2 \left( rR^2 + 2\bar{L} \right) rR^2 \sqrt{\frac{2}{n}} + 4(\bar{L}^2 + r^2 R^4) \sqrt{\frac{1}{2n} \log \frac{1}{\delta}}$$

A similar analysis also gives us with the same confidence

$$\mathcal{L}(A^*) - \hat{\mathcal{L}}(A^*; \mathcal{D}) \leq 4(\bar{L}^2 + \|A^*\|_F^2 R^4) \sqrt{\frac{1}{2n} \log \frac{1}{\delta}}$$

However, an argument similar to that used in the proof of Theorem 1 shows us that

$$\mathcal{L}(\hat{A}) \leq \mathcal{L}(A^*) + C \left( \bar{L}^2 + \left( r^2 + \|A^*\|_F^2 \right) R^4 \right) \sqrt{\frac{1}{n} \log \frac{1}{\delta}}$$

Combining the above inequalities yields the desired result. $\square$

# B  Experiments

In this section we present detailed experimental results, as well as descriptions of evaluation metrics that could not be included in the main text due to lack of space.

## B.1  Evaluation Metrics

We used two metrics to evaluate algorithms in our experiments. Both have been widely adopted in several XML and ranking tasks.

**Precision @ $k$**: this metric has been widely adopted as the metric of choice for evaluating XML algorithms and is motivated by real world application scenarios such as tagging and recommendation where only accuracy at the top of the ranked/recommendation list matters. Formally, the precision at $k$ for a predicted score vector $\hat{\mathbf{y}} \in \mathcal{R}^L$ is the fraction of correct positive predictions in the top $k$ scores of $\hat{\mathbf{y}}$. We sort the labels according to the scores assigned to them by $\hat{\mathbf{y}}$ and then count the number of positive predictions in the top $k$ positions in this ranked list.

**nDCG Evaluation Metric**: Let $S_n$ denote the symmetric group of the set of all permutations of $\{1, 2, \ldots, L\}$. Given a ground truth label vector $\mathbf{y} \in \{0, 1\}^L$, we can definte, for any permutation $\sigma \in S_L$, the Discounted Cumulative Gain (DCG) at k of $\sigma$ as

$$\text{DCG@}k(\sigma, \mathbf{y}) := \sum_{l=1}^{k} \frac{\mathbf{y}_{\sigma(l)}}{\log(l+1)}$$

The *normalized* version of this metric simply divides this by the largest possible DCG@$k$ value over all permutations.

$$\text{nDCG@}k(\sigma, \mathbf{y}) := I_k(\mathbf{y}) \cdot \sum_{l=1}^{k} \frac{\mathbf{y}_{\sigma(l)}}{\log(l+1)},$$

where

$$I_k(\mathbf{y}) := \left( \sum_{l=1}^{\min(k, \|\mathbf{y}\|_0)} \frac{1}{1+l} \right)^{-1},$$

where $\|\mathbf{y}\|_0$ simply counts the number of active labels in the ground truth vector $\mathbf{y}$. Given this, we can now define the nDCG@$k$ loss for score vectors over labels as well. For any score vector $\hat{\mathbf{y}} \in \mathbb{R}^L$, let $\sigma_{\hat{\mathbf{y}}} \in S_L$ denote the corresponding permutation induced on the labels by sorting them in descending order of the scores. Then we define

$$\text{nDCG@}k(\hat{\mathbf{y}}, \mathbf{y}) := \text{nDCG@}k(\sigma_{\hat{\mathbf{y}}}, \mathbf{y}).$$

## B.2  Supplementary Results

Table 2: Data set Statistics: $n$ and $m$ are the number of training and test points respectively, $d$ and $L$ are the number of features and labels, respectively, and $\bar{d}$ and $\bar{L}$ are the average number of nonzero features and positive labels in an instance, respectively.

| Data set | $d$ | $L$ | $n$ | $m$ | $\bar{d}$ | $\bar{L}$ |
|---|---|---|---|---|---|---|
| MediaMill | 120 | 101 | 30993 | 12914 | 120.00 | 4.38 |
| BibTeX | 1836 | 159 | 4880 | 2515 | 68.74 | 2.40 |
| Delicious | 500 | 983 | 12920 | 3185 | 18.17 | 19.03 |
| EURLex | 5000 | 3993 | 15539 | 3809 | 236.69 | 5.31 |
| Wiki10 | 101938 | 30938 | 14146 | 6616 | 673.45 | 18.64 |
| DeliciousLarge | 782585 | 205443 | 196606 | 100095 | 301.17 | 75.54 |
| WikiLSHTC | 1617899 | 325056 | 1778351 | 587084 | 42.15 | 3.19 |
| Amazon | 135909 | 670091 | 490449 | 153025 | 75.68 | 5.45 |
| Ads1M | 164592 | 1082898 | 3917928 | 1563137 | 9.01 | 1.96 |

Table 3: Results on Small Scale data sets : Comparison of precision accuracies of SLEEC with competing baseline methods on small scale data sets. The results reported are average precision values along with standard deviations over 10 random train-test split for each Data set. SLEEC outperforms all baseline methods on all data sets (except Delicious, where it is ranked $2^{nd}$ after FastXML)

| Data set | | Proposed | Embedding | | | | | Tree Based | | | Other | |
|---|---|---|---|---|---|---|---|---|---|---|---|---|
| | | SLEEC | LEML | WSABIE | CPLST | CS | ML-CSSP | FastXML-1 | FastXML | LPSR | OneVsAll | KNN |
| Bibtex | P@1 | **65.57 ±0.65** | 62.53±0.69 | 54.77±0.68 | 62.38 ±0.42 | 58.87 ±0.64 | 44.98 ±0.08 | 37.62 ±0.91 | 63.73±0.67 | 62.09±0.73 | 61.83 ±0.77 | 57.00 ±0.85 |
| | P@3 | **40.02 ±0.39** | 38.40 ±0.47 | 32.38 ±0.26 | 37.83 ±0.52 | 33.53 ±0.44 | 30.42 ±2.37 | 24.62 ±0.68 | 39.00 ±0.57 | 36.69 ±0.49 | 36.44 ±0.38 | 36.32 ±0.47 |
| | P@5 | **29.30 ±0.32** | 28.21 ±0.29 | 23.98 ±0.18 | 27.62 ±0.28 | 23.72 ±0.28 | 23.53 ±1.21 | 21.92 ±0.65 | 28.54 ±0.38 | 26.58 ±0.38 | 26.46 ±0.26 | 28.12 ±0.39 |
| Delicious | P@1 | 68.42 ±0.53 | 65.66 ±0.97 | 64.12 ±0.77 | 65.31 ±0.79 | 61.35 ±0.77 | 63.03 ±1.10 | 55.34 ±0.92 | **69.44 ±0.58** | 65.00±0.77 | 65.01 ±0.73 | 64.95 ±0.68 |
| | P@3 | 61.83 ±0.59 | 60.54 ±0.44 | 58.13 ±0.58 | 59.84 ±0.5 | 56.45 ±0.62 | 56.26 ±1.18 | 50.69 ±0.58 | **63.62 ±0.75** | 58.97 ±0.65 | 58.90 ±0.60 | 58.90 ±0.70 |
| | P@5 | 56.80 ±0.54 | 56.08 ±0.56 | 53.64 ±0.55 | 55.31 ±0.52 | 52.06 ±0.58 | 50.15 ±1.57 | 45.99 ±0.37 | **59.10 ±0.65** | 53.46 ±0.46 | 53.26 ±0.57 | 54.12 ±0.57 |
| MediaMill | P@1 | **87.09±0.33** | 84.00±0.30 | 81.29 ±1.70 | 83.34 ±0.45 | 83.82 ±0.36 | 78.94 ±10.1 | 61.14±0.49 | 84.24 ±0.27 | 83.57 ±0.26 | 83.57 ±0.25 | 83.46 ±0.19 |
| | P@3 | **72.44 ±0.30** | 67.19 ±0.29 | 64.74 ±0.67 | 66.17 ±0.39 | 67.31 ±0.17 | 60.93 ±8.5 | 53.37 ±0.30 | 67.39 ±0.20 | 65.78 ±0.22 | 65.50 ±0.23 | 67.91 ±0.23 |
| | P@5 | **58.45 ±0.34** | 52.80 ±0.17 | 49.82 ±0.71 | 51.45 ±0.37 | 52.80 ±0.18 | 44.27 ±4.8 | 48.39 ±0.19 | 53.14 ±0.18 | 49.97 ±0.48 | 48.57 ±0.56 | 54.24 ±0.21 |
| EurLEX | P@1 | **80.17 ±0.86** | 61.28±1.33 | 70.87 ±1.11 | 69.93±0.90 | 60.18 ±1.70 | 56.84±1.5 | 49.18 ±0.55 | 68.69 ±1.63 | 73.01 ±1.4 | 74.96 ±1.04 | 77.2 ±0.79 |
| | P@3 | **65.39 ±0.88** | 48.66 ±0.74 | 56.62 ±0.67 | 56.18 ±0.66 | 48.01 ±1.90 | 45.4 ±0.94 | 42.72 ±0.51 | 57.73 ±1.58 | 60.36 ±0.56 | 62.92 ±0.53 | 61.46 ±0.96 |
| | P@5 | **53.75 ±0.80** | 39.91 ±0.68 | 46.20 ±0.55 | 45.74 ±0.42 | 38.46 ±1.48 | 35.84 ±0.74 | 37.35 ±0.42 | 48.00 ±1.40 | 50.46 ±0.50 | 53.42 ±0.37 | 50.45 ±0.64 |

Table 4: Stability of SLEEC learners. We show mean precision values over 10 runs of SLEEC on WikiLSHTC with varying number of learners. Each individual learner as well as ensemble of SLEEC learners was found to be extremely stable with with standard deviation ranging from 0.16% on P1 to 0.11% on P5.

| # Learners | 1 | 2 | 3 | 4 | 5 | 6 | 7 | 8 | 9 | 10 |
|---|---|---|---|---|---|---|---|---|---|---|
| P@1 | 46.04 ±0.1659 | 50.04 ±0.0662 | 51.65 ±0.074 | 52.62 ±0.0878 | 53.28 ±0.0379 | 53.63 ±0.083 | 54.03 ±0.0757 | 54.28 ±0.0699 | 54.44 ±0.048 | 54.69 ±0.035 |
| P@3 | 26.15 ±0.1359 | 29.32 ±0.0638 | 30.70 ±0.052 | 31.55 ±0.067 | 32.14 ±0.0351 | 32.48 ±0.0728 | 32.82 ±0.0694 | 33.07 ±0.0503 | 33.24 ±0.023 | 33.45 ±0.0127 |
| P@5 | 18.14 ±0.1045 | 20.58 ±0.0517 | 21.68 ±0.0398 | 22.36 ±0.0501 | 22.85 ±0.0179 | 23.12 ±0.0525 | 23.4 ±0.0531 | 23.60 ±0.0369 | 23.74 ±0.0172 | 23.92 ±0.0115 |

Table 5: nDCG Large-scale data sets : Our proposed method SLEEC is as much as 35% more accurate in terms of nDCG@1 and 33% in terms of nDCG@5 than LEML, a leading embedding method. Other embedding based methods do not scale to the large-scale data sets; SLEEC is also 5% more accurate (w.r.t. nDCG@1 and nDCG@5) than FastXML, a state-of-the-art tree method. '-' indicates LEML could not be run with the standard resources.

| Data set | | SLEEC | LEML | FastXML | LPSR-NB |
|---|---|---|---|---|---|
| Wiki10 | nDCG@1 | **86.14** | 73.47 | 81.71 | 72.72 |
| | nDCG@3 | **76.11** | 64.92 | 69.12 | 61.71 |
| | nDCG@5 | **68.15** | 58.69 | 61.46 | 54.63 |
| Delicious-Large | nDCG1 | **47.53** | 40.73 | 43.34 | 18.59 |
| | nDCG3 | **43.33** | 38.44 | 39.78 | 16.17 |
| | nDCG5 | **41.19** | 37.01 | 37.80 | 15.13 |
| WikiLSHTC | nDCG@1 | **54.81** | 19.82 | 49.37 | 27.43 |
| | nDCG@3 | **47.23** | 14.52 | 44.89 | 23.04 |
| | nDCG@5 | **46.14** | 13.73 | 44.43 | 22.54 |
| Amazon | nDCG@1 | **34.77** | 8.13 | 34.24 | 28.65 |
| | nDCG@3 | **32.74** | 7.30 | 32.09 | 26.40 |
| | nDCG@5 | **31.53** | 6.85 | 30.48 | 25.03 |
| Ads-1m | nDCG@1 | 21.75 | - | **23.31** | 17.95 |
| | nDCG@3 | 23.67 | - | **24.06** | 19.50 |
| | nDCG@5 | **25.06** | - | 24.71 | 20.65 |

Figure 3: Variation of precision accuracy with model size on Ads-1m Data set

Table 6: nDCG Results on Small Scale data sets : Comparison of normalized Discounted Cumulative Gain (nDCG) performance of SLEEC with competing baseline methods on small scale data sets. SLEEC outperforms all baseline methods on all data sets (except Delicious, where it is ranked $2^{nd}$ after FastXML)

| Data set | | Proposed | Embedding | | | | | Tree Based | | | Other | |
|---|---|---|---|---|---|---|---|---|---|---|---|---|
| | | SLEEC | LEML | WSABIE | CPLST | CS | ML-CSSP | FastXML-1 | FastXML | LPSR | OneVsAll | KNN |
| Bibtex | nDCG@1 | **64.49** | 63.10 | 55.03 | 61.99 | 59.60 | 56.86 | 46.04 | 63.78 | 62.98 | 62.62 | 57.81 |
| | nDCG@3 | **59.90** | 58.84 | 50.26 | 57.66 | 53.08 | 52.54 | 40.55 | 59.73 | 57.11 | 59.44 | 52.36 |
| | nDCG@5 | **62.29** | 61.06 | 52.33 | 59.71 | 53.74 | 54.81 | 40.73 | 61.72 | 58.76 | 61.73 | 54.57 |
| Delicious | nDCG@1 | 67.41 | 64.96 | 63.67 | 65.65 | 61.26 | 63.96 | 57.30 | **69.92** | 64.46 | 64.9 | 64.8 |
| | nDCG@3 | 62.46 | 61.80 | 59.47 | 61.52 | 57.85 | 59.07 | 52.97 | **65.91** | 60.47 | 60.94 | 60.71 |
| | nDCG@5 | 59.02 | 58.42 | 56.37 | 58.00 | 54.44 | 54.86 | 49.89 | **62.20** | 56.19 | 56.54 | 57.02 |
| MediaMill | nDCG@1 | **86.61** | 83.99 | 79.86 | 83.79 | 83.97 | 83.21 | 82.11 | 83.73 | 83.65 | 83.67 | 82.59 |
| | nDCG@3 | **80.04** | 75.23 | 72.51 | 74.44 | 75.29 | 72.67 | 73.21 | 74.84 | 74.12 | 73.90 | 72.76 |
| | nDCG@5 | **77.71** | 71.96 | 69.34 | 70.49 | 71.99 | 65.05 | 69.94 | 71.66 | 69.12 | 67.75 | 72.76 |
| EurLEX | nDCG@1 | **79.86** | 61.64 | 68.54 | 70.17 | 58.51 | 69.04 | 47.18 | 70.75 | 74.29 | 75.97 | 77.00 |
| | nDCG@3 | **67.96** | 52.70 | 58.43 | 59.64 | 48.66 | 55.95 | 40.77 | 61.41 | 64.93 | 67.28 | 65.61 |
| | nDCG@5 | 61.59 | 47.75 | 53.02 | 53.79 | 40.79 | 48.17 | 36.09 | 56.05 | 59.43 | **62.54** | 59.39 |

Figure 4: Variation of precision accuracy with model size on Amazon Data set

Figure 5: Variation of precision accuracy with model size on Delicious-Large Data set

Figure 6: Variation of precision accuracy with model size on Wiki10 Data set

Figure 7: Variation of precision accuracy with model size on WikiLSHTC Data set