[Reviews · NeurIPS 2015]

Submitted by Assigned_Reviewer_1

The authors present "X1", a novel kNN classifier using embeddings. The key distinction is that the embeddings are learned to preserve NN relationships in label space, i.e. training points that have similar label sets should be close in the embedded space. (This seems only suited to the mult-label setting - in classification, this would just be grouping points into sets by their labels.) The embedding objective is not penalized for errors on pairs of points that are not nearest neighbors. At test time, the test point is embedded and the labels of kNN points from the training set are used to make a prediction. To speed up the method, the training set is divided into subsets and embeddings are learned for each subset; test points are first bucketed before running inference.

Paper strengths:

+ The paper is well written and easy to read.

+ The authors do a good job motivating the move away from globally low-rank assumptions: while low-rank is convenient for practical purposes, the authors show empirically that these assumptions are violated in practice.

+ The algorithm is explained quite thoroughly, with diligent effort to relate to previous work and to explain the contribution of each of their results. Similar ideas behind the various components of their algorithm have been used in other contexts, but they weave them together quite effectively; it seems like a very practical approach.

+ I'm not too familiar with the particular benchmark task, but they do compare to recent state-of-the-art results very favorably.

+ They provide some interesting theoretical analysis of their method, which other practical methods lack.

Paper weaknesses/questions:

- I didn't really understand the presentation of SVP, but I think for people more familiar with the method it might be enough.

- My main substantial question is about the objective.

Since the projection in (3) is applied after the reconstruction of the inner product, the error for any points that are not nearest neighbors is always zero, correct? What's the intuition for why this leads to a good reproduction of the set of nearest neighbors? On the surface it seems that wildly inaccurate predictions for non-nearest neighbor points could lead to false positives without affecting the objective, as long as the inner products for nearest neighbors are faithfully reconstructed. When querying the training set, these inaccurate inner product values would be retrieved instead...According to the main text, this should be addressed in Theorem 7 in the Appendix, but I would much rather see this discussed in the main paper.
Summary: This is a solid paper with nice experiments and theoretical analysis. I'm still a little unsure about the intuition behind the objective, but the paper has good writing and attention to detail.

Submitted by Assigned_Reviewer_2

The submission describes a scalable method for large multi-label prediction problems with very long-tailed label distributions.

The method first learns a nonlinear embedding of the label (output) vectors that preserves local distances.

A linear map is then learned from features (inputs) into this embedding space.

At test time, the linear map is evaluated to predict a point in the embedding space, and then the labels are predicted via KNN in this space (i.e., the predicted label vector is the sum of those of training examples with similar embeddings).

A few tricks are used to achieve scalability and good generalization: namely, clustering is used to reduce the complexity of KNN, and averaging over the randomness of clustering results is used to mitigate the instability of high-dimensional clustering.

Experiments show that this method is faster to evaluate, more scalable at training, and generalizes better than competing methods.

Significance: The experimental results are the strongest part of the paper, which show convincingly better results in multiple aspects compared to existing state-of-the-art methods.

However, the method is a bit complex compared to other approaches, in the sense that it is a pipeline with a few steps and a number of parameters to tune.

This makes it a bit less likely to catch on and have a big practical impact.

Originality: The basic solution approach builds on previous embedding-based methods, and as such, the component parts have mostly appeared elsewhere previously.

The most novel part about the method seems to be the idea to use a nonlinear, local-distance-preserving embedding as opposed to a linear, global embedding of the labels, but this is a fairly straightforward innovation.

Clarity: Much can be done to improve the clarity of the submission.

First, the paper suffers from information overload in a few places, making it difficult to tell what is important and what is not.

As mentioned elsewhere, the theoretical analysis, though interesting, seems to be of little relevance to the actual method, so perhaps this section should be abridged.

The experiments are very thorough, which is good, but in this section and throughout the paper, I would have preferred it if the discussion had focused on discussing the most important points well rather than trying to cram in as many points as possible.

Quality: Overall, this is a well-thought-out method to solve a real-world problem well.

Thoughtful concern is given to the many details required to make a system performant and scalable on real data.

The experiments are also thorough with strong results.

The theoretical analysis also seems very sophisticated, but unfortunately seems like a tangent with little relevance to the rest of the work.

The weakest part of the work is that the novelty of the method is relatively low.

Other comments:

From a technical perspective, I was a bit disappointed at the disconnect between the proposed optimization for the embedding and the method used to solve it.

The idea of learning an embedding that is nonlinear as a map from labels, but linear as a map from features is very interesting, because the linearity-from-features constraint serves to regularize an embedding which might otherwise be too powerful to produce meaningful results.

Unfortunately, the algorithm used to actually optimize this objective learns the embedding completely ignoring the constraint that it must be realizable as a linear map from the features.

So, the obtained solution effectively optimizes an objective which differs from the desired objective (3) in a very important way.

This is also one reason why the ERM bound fails to provide justification for the method: there is little reason to believe that the obtained solution should be close to the optimum of (3).

A related concern is that preserving local distances alone is not sufficient to ensure good accuracy of KNN, since a nonlinear embedding can easily preserve local distances while also pushing points into neighborhoods in which they do not belong.

This is why methods such as Large Margin Nearest Neighbor exist.

Unfortunately, such methods tend to be computationally expensive, so I worry that taking this into account would spoil the computational efficiency of the method.

These concerns make it a little surprising that the method works so well in practice.

POST-REBUTTAL COMMENTS:

The rebuttal did not significantly change my opinion.

I am positive overall about the work, mostly due to the experimental results, but I still have a few concerns.

Basically, the presentation makes it seem as though certain decisions are motivated by rigorous theory, but this is not really the case.

The two-step optimization procedure used in practice differs in a critically important way from the theoretically well-motivated joint optimization.

The fact that the objective does not properly preserve neighborhoods also makes it surprising that the method works well in practice.

If I were to revise the manuscript, I would put less emphasis on the marginally-related theoretical results and more emphasis on giving plausible explanations, empirical or otherwise, for these counterintuitive phenomena.
Summary: The submission describes a scalable method for multi-label prediction problems with a large number of labels.

Experimental results are thorough and show a significant advantage over other methods, but the method is a bit involved and arguably not that original in the sense that it mostly combines tricks from other methods.

Submitted by Assigned_Reviewer_3

Summary:

This paper considers the problem of extreme multi-label learning: multi-label classification with a large number of labels. Authors' identify the main drawback of existing emedding based approaches (e.g., compressed sensing, output coding) -- their low-rank assumption does not hold in real-world data -- and propose a method to overcome it. The key idea is to learn a ensemble of local distance preserving embeddings be able to accurately predict the rare labels (source of the above drawback). The proposed method achieves fast training/prediction by performing K-NN classification within the relevant cluster. Experiments are performed on multiple large-scale benchmark corpora with good training time, testing time, and accuracy results.

Pros:

- Studies a very important and timely problem with large number of applications. - Good problem formulation; and proposed solution is well-motivated along with bounds on generalization. - Mostly well-written paper. - Good experimental results.

Cons:

- Missing evaluation with task losses other than Precision@K

Detailed Comments:

- Authors' perform evaluation with "precision@K" loss function. In the multi-label classification literature, several other task losses (e.g., Micro/Macro F1, Hamming accuracy) are regularly used. I would have liked to see results with other loss functions. When the number of relevant labels (the labels that take value "1") is very large, precision@K may not be the best way to measure the performance of the predictor.
Summary: Summary:

This paper considers the problem of extreme multi-label learning: multi-label classification with a large number of labels. Authors' identify the main drawback of existing emedding based approaches (e.g., compressed sensing, output coding) -- their low-rank assumption does not hold in real-world data -- and propose a method to overcome it. The key idea is to learn a ensemble of local distance preserving embeddings be able to accurately predict the rare labels (source of the above drawback). The proposed method achieves fast training/prediction by performing K-NN classification within the relevant cluster. Experiments are performed on multiple large-scale benchmark corpora with good training time, testing time, and accuracy results.

Author Feedback
Author rebuttal: We thank all the reviewers for their helpful feedback. We would especially like to thank the area chair for the meta review at this stage.

Clarity of the Paper:
While two of the reviewers have found the paper well motivated and easy to read, we will make greater efforts to improve the presentation, especially that of the description of SVP, theoretical analysis and experiments. Due to multi-faceted nature of the problem, we needed to highlight all aspects of the problem and our solution (in the Introduction and Experiments sections) to convince that our solution is indeed practical and considers the problem holistically. To clarify, the algorithm has been named X1 (XONE) which stands for eXtreme lOcally Non-linear Embeddings.

Objective function and Preserving Local neighbourhood :
A common concern raised by Reviewers 1 and 4 is whether preserving local distances is sufficient to prevent non-neighbours from entering the immediate neighbourhood. We would like to point out that for all datasets, we have adopted large neighbourhood sizes (e.g. for WikiLSHTC we selected a neighbourhood size of 15 out of which around 10 documents do not share more than a single label). This larger neighbourhood ensures that we preserve distances to a few far away points in our objective function as well, thus prohibiting non-neighbouring points from entering the immediate neighbourhood of typical points. Further, we did try out LMNN and as pointed out by reviewer 4: the computational cost was high, but with negligible benefits in terms of accuracy.

Loss functions other than Prec@k (Reviewer 3):
The key differences between standard and extreme multi-label classification settings lie in the statistics of the datasets and metrics used to evaluate performance. Standard measures of performance such as Hamming loss, Macro/Micro F1 do not directly apply in the extreme setting because :
a. In most real world applications of extreme multi-label classification like ad-suggestion and document tagging, only k slots are available for prediction and therefore it is more important to predict the top k relevant labels correctly rather than all of them
b. In extreme learning, labels are often missing since no expert can go through the entire set of relevant labels even for a single data point. As such, a 0 in our ground truth does not mean that a label is irrelevant as it could simply be a missing label. Standard measures such as Hamming loss or Micro/Macro F1 would treat such labels as irrelevant and therefore produce biased results. Metrics such as precision and NDCG which are computed only on positive labels are therefore more suitable. We will present results on NDCG as well so as to include evaluation on multiple metrics.

Originality of our method (Reviewer 4):
Our contribution towards the extreme multi-label learning problem is twofold - the insight that non-linear embeddings are a key step is in itself a fairly non-trivial contribution, given that several published papers in this area so far have worked with linear embeddings only, and one that requires analysis of large datasets. Furthermore, learning non-linear embeddings at such large scales is a challenging task and required careful design of the method.

Optimization (Reviewer 4):
We do indeed optimize our objective function using a two step procedure: first learning a non-linear map from the label space to the embedded space and then learning a linear map from the feature space to the embeddings. We did try the joint optimization of the objective, as pointed out by Reviewer 4. However, we found that the resulting non-convex optimization extremely challenging to perform at our scales. Note that the high data dimensionality renders the joint optimization over the d x \hat L matrix W infeasible since each step of a projected gradient style method becomes computationally expensive. Our method overcomes this by first optimizing over the embedding matrix (nc x \hat L, nc : number of points in a cluster << data dimensionality) and then learning the sparse linear regressors.

Complexity of our Method :
Having implemented all existing techniques in the literature to form the base lines, we feel that our proposed technique has more or less the same level of complexity in comparison. Nevertheless, we will make the code for our technique publicly available so that it is easy for others to build upon our technique.